# T-cell responses to sequentially emerging viral escape mutants shape long-term HIV-1 population dynamics

Tomohiro Akahoshi[1], Hiroyuki Gatanaga[2,3], Nozomi Kuse[1,2], Takayuki Chikata[1,2], Madoka Koyanagi [1], Naoki Ishizuka[4], Chanson J. Brumme [5,6], Hayato Murakoshi[1,2], Zabrina L. Brumme [5,7], Shinichi Oka[2,3], Masafumi Takiguchi [1,2]*

1 Center for AIDS Research, Kumamoto University, Kumamoto, Japan, 2 Division of International Collaboration Research, Joint Research Center for Human Retrovirus Infection, Kumamoto University, Tokyo, Japan, 3 AIDS Clinical Center, National Center for Global Health and Medicine, Tokyo, Japan, 4 Cancer Institute Hospital of JFCR, Tokyo, Japan, 5 British Columbia Centre for Excellence in HIV/AIDS, Vancouver, Canada, 6 Department of Medicine, University of British Columbia, Vancouver, Canada, 7 Faculty of Health Sciences, Simon Fraser University, Burnaby, Canada

* masafumi@kumamoto-u.ac.jp

**Data Availability Statement:** All relevant data are within the manuscript and its Supporting Information files.

## Abstract

HIV-1 strains harboring immune escape mutations can persist in circulation, but the impact of selection by multiple HLA alleles on population HIV-1 dynamics remains unclear. In Japan, HIV-1 Reverse Transcriptase codon 135 (RT135) is under strong immune pressure by HLA-B*51:01-restricted and HLA-B*52:01-restricted T cells that target a key epitope in this region (TI8; spanning RT codons 128–135). Major population-level shifts have occurred at HIV-1 RT135 during the Japanese epidemic, which first affected hemophiliacs (via imported contaminated blood products) and subsequently non-hemophiliacs (via domestic transmission). Specifically, threonine accumulated at RT135 (RT135T) in hemophiliac and non-hemophiliac HLA-B*51:01+ individuals diagnosed before 1997, but since then RT135T has markedly declined while RT135L has increased among non-hemophiliac individuals. We demonstrated that RT135V selection by HLA-B*52:01-restricted TI8-specific T-cells led to the creation of a new HLA-C*12:02-restricted epitope TN9-8V. We further showed that TN9-8V-specific HLA-C*12:02-restricted T cells selected RT135L while TN9-8T-specific HLA-C*12:02-restricted T cells suppressed replication of the RT135T variant. Thus, population-level accumulation of the RT135L mutation over time in Japan can be explained by initial targeting of the TI8 epitope by HLA-B*52:01-restricted T-cells, followed by targeting of the resulting escape mutant by HLA-C*12:02-restricted T-cells. We further demonstrate that this phenomenon is particular to Japan, where the HLA-B*52:01-C*12:02 haplotype is common: RT135L did not accumulate over a 15-year longitudinal analysis of HIV sequences in British Columbia, Canada, where this haplotype is rare. Together, our observations reveal that T-cell responses to sequentially emerging viral escape mutants can shape long-term HIV-1 population dynamics in a host population-specific manner.

**Funding:** This research was supported in part by a grant-in-aid for AIDS Research (15fk0410019h0001, 16fk0410202h0002, and 17fk0410302h0003) from AMED to MT and by a grant-in-aid (26293240 and 26860335) for scientific research from the Ministry of Education, Science, Sports and Culture, Japan, to MT and TA, and by a project grant from the Canadian Institutes of Health Research (PJT-148621) to ZLB. ZLB is supported by a Scholar Award from the Michael Smith Foundation for Health Research. The funders had no role in study design, data collection and analysis, decision to publish, or preparation of the manuscript.

**Competing interests:** The authors have declared that no competing interests exist.

## Author summary

HIV-1 strains harboring immune escape mutations can accumulate in circulation, but it remains unclear to what extent these 'escaped' HIV-1 strains continue to evolve under ongoing population-level immune pressures. We investigated population-level changes at HIV-1 Reverse Transcriptase codon 135 (RT135), which is under pressure by T cells restricted by HLA-B*51:01 and B*52:01, highly frequent alleles in Japan. While threonine initially accumulated at RT135, RT135L has subsequently increased markedly. Our findings revealed that RT135V selection by HLA-B*52:01-restricted T-cells led to the creation of a new epitope restricted by HLA-C*12:02, an allele in strong linkage disequilibrium with HLA-B*52:01. HLA-C*12:02-restricted T cells in turn suppressed replication of RT135T virus and selected RT135L. Notably, population-level shifts at this codon are particular to Japan, where HLA-B*52:01-C*12:02 represents the most prevalent HLA haplotype. Our findings highlight multiple virus-specific T cells as dynamic drivers of population-level – and host population-specific – HIV-1 evolution over the long term.

## Introduction

HIV-1 mutations, many of which are selected by HIV-1-specific cytotoxic T lymphocytes (CTLs), can accumulate at the population level [1–5]. Some of these mutations may revert to wild-type upon transmission if the new hosts cannot elicit epitope-specific CTLs and/or if these mutant viruses have reduced viral fitness [6–9]. Many mutations however do not carry a fitness cost, and can therefore persist and accumulate [3,10–12]. Studies of HLA-associated polymorphisms in HIV-1 suggest that most of these are selected by CTLs [13–17], and a recent report that compared historic and recent HIV-1 strains in North America demonstrated that, on average, circulating HLA-associated mutation frequencies have approximately doubled since the 1980s [18,19]. Thus, at the population level, HIV-1 strains are continually evolving to adapt to antiviral HLA-mediated immune responses in their host population(s).

As immune escape mutations compromise the recognition of HIV-1 by virus-specific T cells, their accumulation at the population level compromises T cell-mediated control of HIV-1 [10,20–23]. These negative implications also extend to protective HLA alleles such as HLA-B57, HLA-B27, and HLA-B51 [6,24–28]. For example, the RT I135T mutation selected by HLA-B*51:01-restricted T cells specific for the TI8 epitope (RT codons 128–135) completely abrogates HIV-1 recognition by these cells [2,29]. Consequently, although HLA-B*51:01 was a protective allele in Japan more than 30 years ago [29,30], HLA-B*51:01 is no longer protective in individuals infected more recently [31,32]. This is likely because the RT135T mutation accumulated to particularly high levels in Japan [2,33].

In Japan however, RT135 mutations also frequently occur in individuals harboring HLA-B*52:01[33], which is common in the region [34]. Specifically, RT135T is predominantly observed in HLA-B*51:01+ individuals whereas RT135T, RT135L, and RT135V are observed at relatively equal frequencies in HLA-B*52:01+ individuals [33]. Curiously however, while HLA-B*51:01-restricted and HLA-B*52:01-restricted TI8-specific CTLs fail to recognize target cells infected with RT135T or RT135L mutant virus, they can still recognize cells infected with RT135V virus [2,33]. The mechanisms underlying differential RT135 mutation patterns in HLA-B*51:01+ and HLA-B*52:01+ individuals however remain unclear.

Two hypotheses are plausible. Undiscovered HLA-B*52:01-restricted epitopes in the region could explain RT135 mutations, but our previous intensive searches did not identify any additional HLA-B*52:01-restricted epitopes in this region that contained RT135 [35]. Alternatively,

T-cells restricted by allele(s) in strong linkage disequilibrium with HLA-B*52:01, that target undiscovered epitopes in the region, could select RT135 mutations. In Japan, HLA-C*12:02 is in very strong linkage disequilibrium (LD) with HLA-B*52:01 (LD = 0.99) [34,36], and the HLA-B*52:01-C*12:02 haplotype, at 20% frequency, represents the most prevalent haplotype in Japan. As such, we hypothesized that HLA-C*12:02-restricted T cells may contribute to the selection of different mutations in individuals carrying this haplotype.

In the present study, we searched for novel HLA-C*12:02-restricted T cell epitopes covering RT135 and investigated the role of these T cells in selecting mutations at RT135 in individuals harboring the HLA-B*52:01-C*12:02 haplotype. We further explored the possibility that such T cells could underpin the dynamic changes in population-level RT135 mutation frequencies that have occurred over the course of the HIV-1 epidemic in Japan.

## Results

### Accumulation of RT135V mutation in HLA-B*52:01⁺HLA-C*12:02⁺ hemophiliacs with HIV

To exclude the effect of ongoing domestic HIV-1 transmission on the accumulation of mutant viruses in the population, we analyzed RT135 amino acid frequencies in 95 Japanese hemophiliacs who acquired HIV-1 via transfusion with contaminated blood products imported from the USA around 1983 [37,38]. Overall, 49% acquired HIV-1 carrying RT135I (wild-type virus), but this percentage rose to 87% when restricting to B*51:01⁻/B*52:01⁻ hemophiliacs (**Fig 1A**). This supports our previous observation that the majority of hemophiliacs acquired wild-type (RT135I) virus [2]. The frequency of RT135V was significantly higher among HLA-B*52:01⁺ individuals than those lacking this allele ($p = 8.9x10^{-5}$, $q = 5.3x10^{-4}$), whereas RT135T was significantly higher among HLA-B*51:01⁺ individuals than those lacking this allele ($p = 3.8X10^{-5}$, $q = 3.4x10^{-4}$) (**Fig 1A**). However, no other RT135 mutations were enriched among HLA-B*52:01⁺ or HLA-B*51:01⁺ individuals. These results clearly demonstrate that RT135V and RT135T mutations were overrepresented among HLA-B*52:01⁺ and HLA-B*51:01⁺ hemophiliacs, respectively.

We next analyzed RT135 mutation frequencies in non-hemophiliac Japanese individuals who had been diagnosed with HIV-1 before 1997, and compared these to the frequencies observed in the hemophiliacs. The results revealed that RT135T had accumulated to comparable levels in HLA-B*51:01⁺ non-hemophiliac and hemophiliac individuals (**Fig 1B**). In contrast, RT135V mutation frequencies were markedly lower in HLA-B*52:01⁺ non-hemophiliac individuals compared to HLA-B*52:01⁺ hemophiliacs (**Fig 1B**). Taken together our results suggest that, not only does RT135 mutation selection differ between HLA-B*52:01⁺ and HLA-B*51:01⁺ individuals, but that it may also differ between HLA-B*52:01⁺ hemophiliac and non-hemophiliac individuals, where RT135 mutant frequencies in the latter group may be further influenced by ongoing domestic HIV-1 transmission over time.

### Generation of novel HLA-C*12:02-restricted epitope after selection of RT135V mutation by HLA-B*52:01-restricted TI8-specific CTLs

Only one nucleotide substitution between I (ata) and T (a**c**a) or V (**g**ta) at RT135 was observed in hemophiliacs and non-hemophiliacs (**S1 Fig** and **S1 Table**), suggesting that RT135T and RT135V each evolved from RT135I. It is not clear however why RT135V predominated in HLA-B*52:01⁺ hemophiliacs while RT135T predominated in HLA-B*51:01⁺ ones. We hypothesized that the differential RT135 mutation patterns observed in HLA-B*51:01⁺ and HLA-B*52:01⁺ individuals may be mediated by HLA-C*12:02-restricted T cell pressures in the

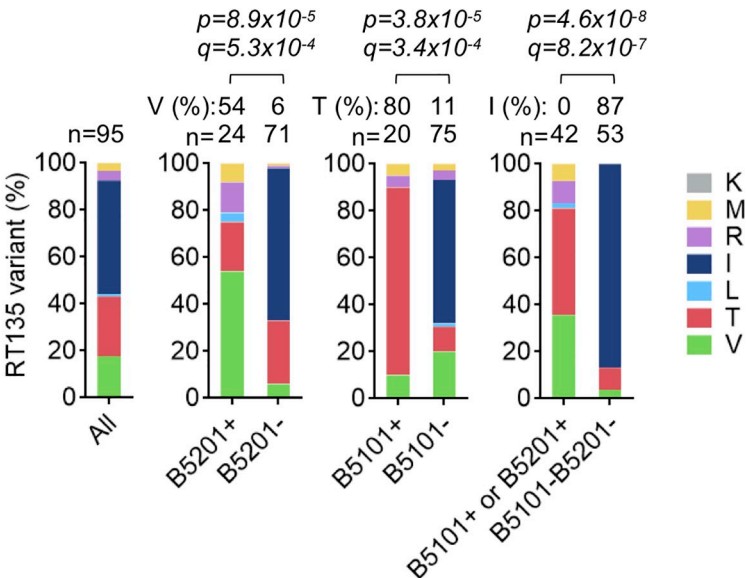

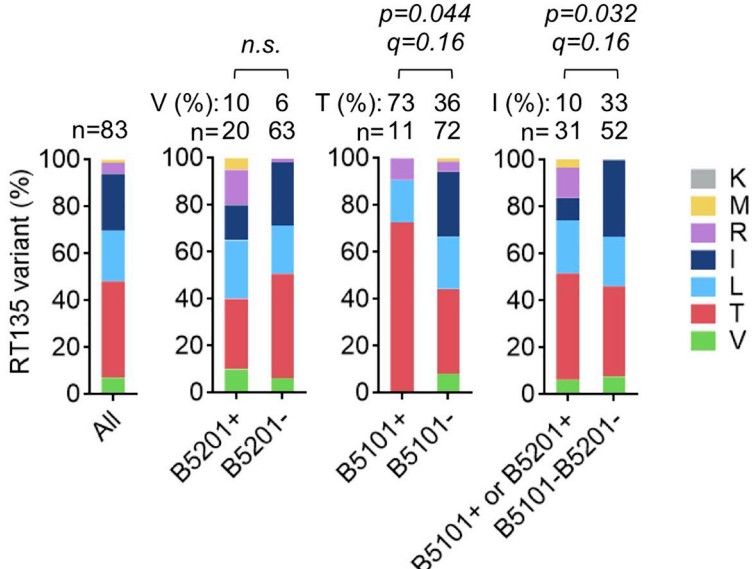

**Fig 1. Amino acid variation at RT135 in Japanese individuals with chronic HIV-1. A**. HIV-1 RT135 amino acid variation in 95 Japanese hemophiliac individuals with HIV-1. RT135 amino acid frequencies were compared between groups using fisher's exact test (two-tailed). Multiple tests were addressed using q values (indicated below the *P* values). **B**. RT135 amino acid variation in 83 Japanese male individuals diagnosed with HIV-1 before 1997. RT135 amino acid frequencies were compared as in Fig 1A.

latter group, since HLA-B*52:01 and HLA-C*12:02 are in strong linkage disequilibrium. We therefore sought to identify novel HLA-C*12:02-restricted CTL epitopes spanning RT135 using a cocktail of overlapping 11-mer peptides including RT135 (Pol cocktail 15). We identified one individual KI-793 who exhibited weak T cell responses to Pol cocktail 15 (**S2A Fig**). We further found that cultured T cells stimulated with KN11 peptide exhibited a weak

HLA-C\*12:02-restricted recognition of KN11 peptide, but no recognition by HLA-B\*52:01-restricted cells (**S2B Fig**). Since this individual harbored RT135V mutant virus, we speculated that his T cells more effectively recognized a peptide containing the RT135V mutation than the wild-type peptide. Indeed, when PBMCs from this individual were cultured after stimulation with KN11 or KN11-10V peptide, the latter induced far stronger responses (**S2C Fig**). We next analyzed the responses of KN11-10V-induced T cells to truncated peptides of KN11-10V. The T cells recognized TN9-8V most effectively (**Fig 2A**), identifying this as the optimal HLA-C\*12:02-restricted epitope. Additional analysis using an HLA-C\*12:02-TN9-8V tetramer revealed a high frequency of HLA-C\*12:02-restricted TN9-8V-specific CD8[+] T cells in PBMCs from KI-793 (1.82% of CD8[+] T cells) (**Figs 2B and S3A**). These results together indicate that HIV-1 strains harboring the RT135V mutant contain a novel HLA-C\*12:02-restricted TN9-8V T cell epitope.

The results further suggested that TN9-8V-specific CD8[+] T cells would be elicited in HLA-B\*52:01[+] HLA-C\*12:02[+] individuals who acquired RT135V virus, but not in those who acquired wild-type virus or other mutants at this position. Indeed, TN9-8V-specific CD8[+] T

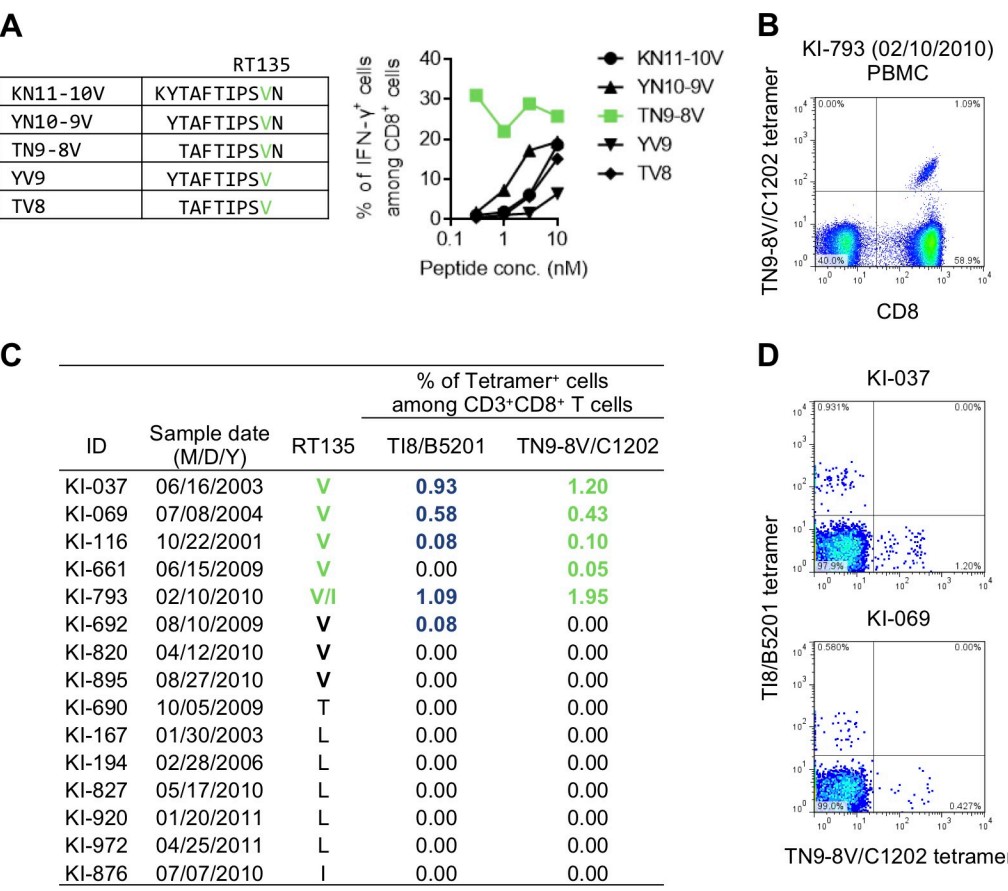

**Fig 2. Identification and characterization of a novel HLA-C\*12:02-restricted epitope TN9-8V. A**. Identification of an optimal HLA-C\*12:02-restricted epitope, TN9-8V. Recognition of KN11-10V-specific bulk CD8[+] T cells from an individual harboring RT135V virus (participant KI-793) by C1R-C1202 cells prepulsed with KN11-10V peptide or its truncated derivatives was analyzed by ICS assay. **B**. Detection of HLA-C\*12:02-restricted TN9-8V-specific CD8[+] T cells in PBMCs from KI-793 by tetramer staining. **C.** Frequencies of HLA-B\*52:01-restricted TI8-specific and HLA-C\*12:02-restricted TN9-8V-specific CD8[+] T cells in HLA-B\*52:01[+]C\*12:02[+] individuals with chronic HIV-1, by staining of TN9-8V/C1202 and TI8/B5201 tetramers. RT135 amino acid variation in these individuals was determined by bulk sequencing. **D**. Representative participants harboring TN9-8V-specific and TI8-specific CD8[+] T cells.

cells were detected in 5 of 8 HLA-B*52:01+HLA-C*12:02+ individuals harboring RT135V mutant virus, whereas these T cells were not detected in any of the 7 HLA-B*52:01+HLA-C*12:02+ individuals harboring wild-type or other mutant virus (Fig 2C). This directly supports the idea that TN9-8V-specific CD8+ T cells are specifically elicited in HLA-B*52:01+HLA-C*12:02+ individuals infected with RT135V virus.

Of note, a previous study showed that TI8-specific HLA-B*52:01-restricted CD8+ T cells suppressed replication of the 8V mutant virus less than that of wild-type virus [33], suggesting that HLA-B*52:01-restricted CD8+ T cells select RT135V. Indeed, TI8-specific and TN9-8V-specific T cells were detected only in individuals harboring RT135V mutant virus, but not in those harboring other mutant viruses (Fig 2C and 2D). This supports the notion that the C*12:02-TN9-8V epitope is "created" following B*52:01-mediated selection of RT135V in vivo.

## Recognition of target cells infected with RT135 mutant viruses by TN9-8V-specific T cells

To clarify the ability of TN9-8V-specific T cells to recognize RT135 variants, we established TN9-8V-specific CD8+ T cell lines from 3 individuals (KI-069, KI-661, and KI-793) and assessed the ability of these T cell lines to recognize four peptides (TN9-8T, -8R, -8L, and -8I) versus the cognate TN9-8V. These T cell lines cross-recognized TN9-8T, though to a lesser extent than TN9-8V (Fig 3A). We measured the binding affinity of TN9-8V and the mutant peptides to HLA-C*12:02 using the HLA class I stabilization assay. Among all peptides tested,

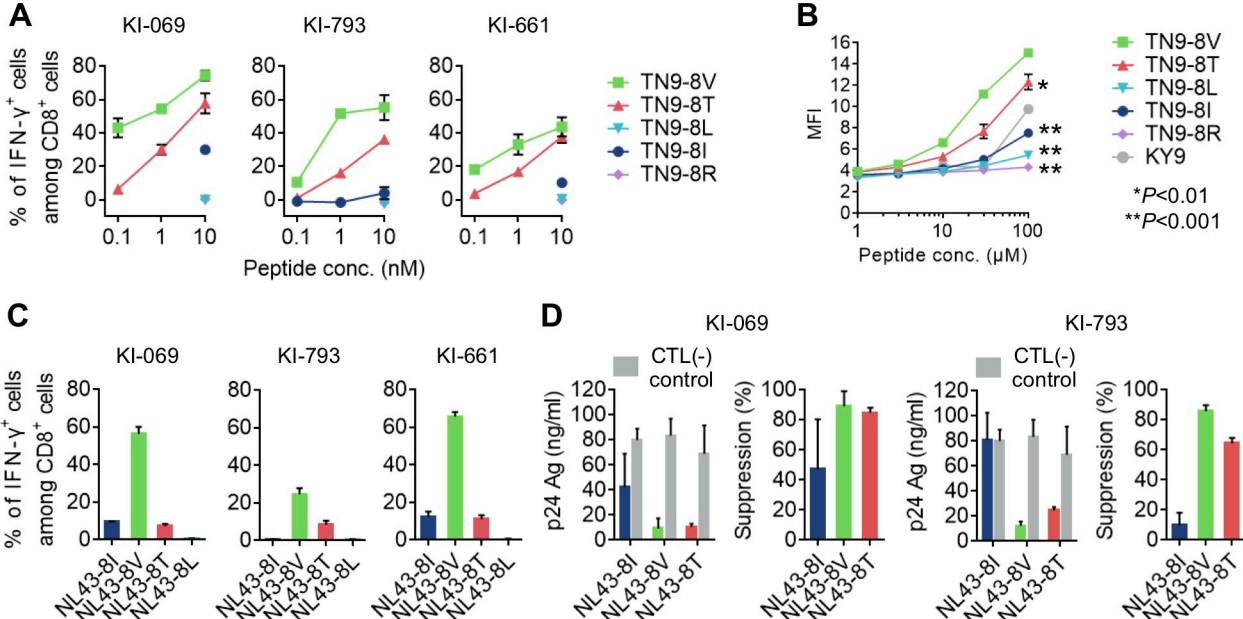

Fig 3. Recognition of RT135 mutant virus-infected cells by HLA-C*12:02-restricted TN9-8V-specific CTL lines. A. Recognition of TN9-8V and its mutant peptides by TN9-8V-specific T cell lines derived from HLA-B*52:01+C*12:02+ individuals with HIV-1. The T cell responses to C1R-C1202 cells prepulsed with TN9-8V or its mutant peptides were analyzed by ICS. B. Binding affinity of TN9-8V and its mutant peptides to HLA-C*12:02 molecules by an HLA class I stabilization assay using RMA-S-C1202 cells. KY9 is a reported HLA-C*12:02-restricted epitope. Statistical analysis was conducted using Student's t-test (two-tailed). Two independent experiments showed the same result. C. Recognition of cells infected with RT135I virus (NL43; NL43-8I) or the mutant virus (NL43-8V, -8T or -8L) by TN9-8V-specific CTL lines. The T-cell responses to 721.221-CD4-C1202 cells infected with NL43-8I or the mutant virus were measured by ICS. The frequencies of p24+ cells among 721.221-CD4-C1202 cells infected with NL43-8I, -8V, -8T, and -8L were 72.5%, 85.6%, 73.6%, and 59.8%, respectively, for KI-069 and KI-661, while they were 63.4%, 67.9%, 64.6%, and 40.4%, respectively, for KI-793. See also S4 Fig. D. Suppression of viral replication by TN9-8V-specific CTL lines, at an E:T ratio of 1:1. The concentration of p24 Ag in the culture supernatants collected on Day 5 was measured by ELISA (left) and the percentage of suppression was calculated (right). Data are shown as the means and ±SD of triplicate assays (A-D).

TN9-8V and TN9-8T exhibited the highest and second highest affinities to HLA-C*12:02, respectively. Of note, these affinities were stronger than those of the known HLA-C*12:02-restricted HIV-1 epitope KY9. In contrast, TN9-8L and TN9-8R did not bind to HLA-C*12:02 (**Fig 3B**), indicating that RT135L/R mutations critically affect peptide recognition by HLA-C*12:02-specific T cells.

We next investigated the ability of these T cell lines to recognize cells infected with the mutant viruses and to suppress HIV-1 replication. All three TN9-8V-specific CD8+ T cell lines recognized 721.221-CD4-C1202 cells infected with RT135V and RT135T mutant virus, though recognition of RT135V was consistently stronger than RT135T. In contrast, recognition of RT135I virus was low or nonexistent (**Figs 3C and S4**). Moreover, while the T cell lines from KI-069 and KI-793 effectively suppressed the replication of both RT135V and RT135T viruses, they only partially suppressed RT135I virus (KI-069) or failed to suppress it (KI-793) (**Fig 3D**). These results together suggest that TN9-8V-specific T cells effectively suppress replication of both RT135V and RT135T viruses *in vivo*.

## Population-level shifts in HIV-1 RT135 mutation frequencies Japan over two decades

Among Japanese hemophiliacs, RT135T was predominantly selected in HLA-B*51:01+ individuals while RT135V was predominantly selected in HLA-B*52:01+ individuals. RT135T was also predominantly selected in HLA-B*51:01+ non-hemophiliacs. Somewhat in contrast however, RT135T, RT135L and RT135V occurred at relatively equal frequencies among HLA-B*52:01+ non-hemophiliacs [33]. These findings suggest that B*52:01-associated selection pressures on RT135 differed between the hemophiliac and non-hemophiliac cohorts. As the HIV-1 epidemic in Japanese non-hemophiliacs is primarily fueled by ongoing domestic transmission, we investigated trends in RT135 frequencies over time.

To do so, we analyzed RT135 mutation frequencies in 1,478 non-hemophilic Japanese individuals diagnosed with HIV-1 between 1997 and 2015. Analysis was performed in the cohort as a whole, and stratified by allele carriage (HLA-B*51:01+B*52:01/C*12:02-, HLA-B*51:01-B*52:01/C*12:02+, and HLA-B*51:01-B*52:01/C*12:02-). RT135 amino acid frequencies were compared across four periods according to year of HIV-1 diagnosis: before 1997 (<1997), 1998–2003, 2004–2009, and 2010–2015 (**Fig 4A**). The frequencies of V, T, R, M, L, K and I changed significantly across these periods, overall and in stratified analyses (all *P<0.001*). Overall, the frequency of wild-type RT135I among non-hemophiliacs declined from 25% before 1997 to 12% in 2010–2015, confirming gradual accumulation of RT135 mutants in the population (*P<0.0001*). Notably, the frequency of RT135L increased during these periods (22% to 44%, *P<0.0001*), whereas that of RT135T decreased (41% to 25%, *P = 0.0005*) (**S2 Table**). Stratification by HLA allele carriage further revealed that RT135L increased in all groups (*P<0.0001*). In contrast, RT135T decreased in only 2 groups: HLA-B*51:01+B*52:01-/C*12:02- individuals and HLA-B*51:01-B*52:01-/C*12:02- individuals (*P = 0.001*) (**S2 Table**).

To demonstrate that these significant increases in RT135L mutation frequencies are particular to Japan, and not mirrored in global regions where B*52:01/C*12:02 carriage is uncommon, we analyzed RT135 amino acid frequencies in individuals with HIV subtype B recruited from 2000 to 2015 in British Columbia (BC), Canada. Of 6,258 individuals analyzed, 56% and 27% harbored RT135I or RT135T, respectively, but only 1% of them harbored RT135L. Moreover, RT135 amino acid frequencies remained unchanged over three periods: 2000–2003, 2004–2009, and 2010–2015 (**Fig 4B**). In British Columbia, population frequency of HLA-B*51:01 is 11.7% whereas that of HLA-B*52:01 or -C*12:02 is only 1.4% [24], suggesting

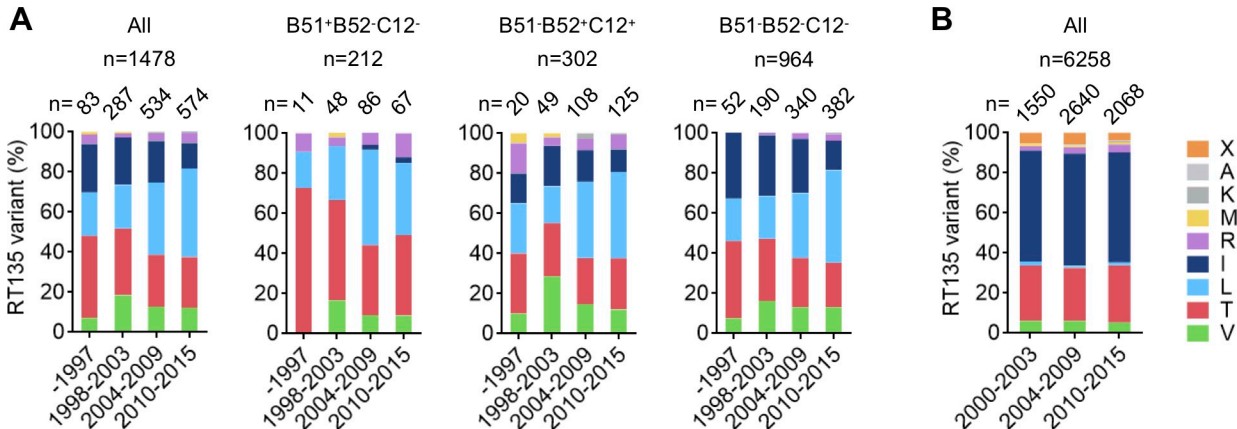

**Fig 4. Amino acid variation at RT135 in Japanese and Canadian individuals with subtype B HIV-1 over 2 decades. A.** HIV-1 RT135 amino acid frequencies in 1,478 Japanese non-hemophiliac individuals with chronic HIV-1 subtype B before 1997, and between 1998–2003, 2004–2009, and 2010–2015. The cohort is analyzed overall, and stratified by HLA allele carriage as follows: HLA-B*51:01⁺B*52:01⁻C*12:02⁻ (n = 212); HLA-B*51:01⁻B*52:01⁺C*12:02⁺ (n = 302); and HLA-B*51:01⁻B*52:01⁻C*12:02⁻ (n = 964). Statistical analysis was conducted using the Cochran-Mantel-Haenszel test (See also S2 Table). **B.** RT135 amino acid frequencies in 6,258 Canadian individuals with chronic HIV-1 subtype B recruited from 2000 to 2015. "X" indicates a mixture of two or more amino acids at that residue (as bulk HIV-1 sequencing was performed).

that RT135V selection by HLA-B*52:01-restricted TI8-specific T cells and elicitation of HLA-C*12:02-restricted TN9-8V-specific T cells would occur only rarely in this population.

## Selection of RT135L by TN9-8V-specific HLA-C*12:02-restricted T cells

Our observation that TN9-8V-specific CD8⁺ T cells could not recognize target cells pulsed with TN9-8L peptide nor cells infected with RT135L mutant virus (Fig 3A and 3C) suggests that RT135L confers escape from these T cells. To investigate this, we analyzed HIV-1 evolution in a hemophiliac individual homozygous for the HLA-B*52:01⁺HLA-C*12:02⁺ haplotype (KI-528). Longitudinal bulk and deep sequencing of plasma HIV-1 RNA revealed that this individual acquired both wild-type (RT135I) and RT135V viruses in May 2003 (Fig 5A). By March 2006, RT135V had disappeared and RT135L had emerged as the dominant variant (68%). Moreover, within-host viral variants harboring RT135L comprised 2 groups having cta and tta at RT135, suggesting that both variants could have evolved from either RT135I or RT135V sequences via only one nucleotide change. By June 2007, RT135L had increased to 92% frequency, with both cta and tta variants in equal proportions.

We next investigated the emergence of TI8-specific HLA-B*52:01-restricted and TN9-8V-specific HLA-C*12:02-restricted CD8⁺ T cells in this individual. Analysis using B*52:01-TI8 and C*12:02-TN9-8V tetramers confirmed the presence of TI8-specific and TN9-8V-specific CD8⁺ T cells in May 2004 (Figs 5B and S3B). The frequency of these T cells however decreased in April 2006, after RT135L had become the dominant variant. We stimulated PBMCs from this individual with TN9-8V peptide and cultured them to induce TN9-8V-specific T cells. The bulk T cells responded to TN9-8V peptide but not TN9-8L peptide (Fig 5C), suggesting that the TN9-8V-specific T cells can select RT135L mutant virus.

## Suppression of RT135T mutant virus by TN9-8T-specific HLA-C*12:02-restricted T cells

The observation that TN9-8T has a strong affinity for HLA-C*12:02 raises the possibility that TN9-8T-specific T cells can target RT135T virus-infected cells. To investigate this, we analyzed an HLA-B*52:01⁺C*12:02⁺ individual, KI-638, who had acquired both RT135I and RT135T

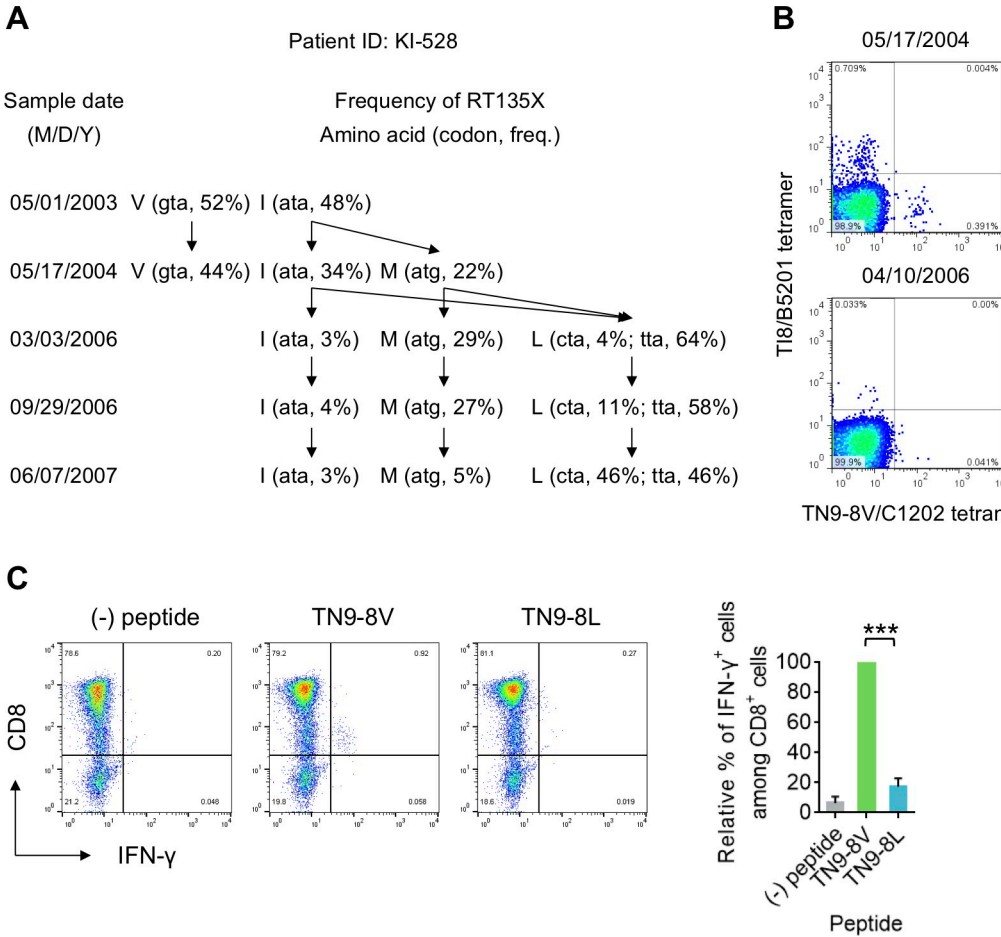

**Fig 5. Longitudinal analysis of RT135 mutant selection in an HLA-B\*52:01⁺C\*12:02⁺ individual. A**. Longitudinal sequence analysis at RT135 in an HLA-B\*52:01⁺C\*12:02⁺ hemophiliac individual with HIV-1, KI-528. Frequencies of amino acids at RT135 were determined by both bulk and deep sequencing. Both analyses showed similar results. The amino acids at RT135, their codon usage, and their frequency are shown at each sampling date. Estimated pathways of viral evolution during May 2003-June 2007 are indicated using arrows. **B**. Detection of TI8- and TN9-8V-specific CD8⁺ T cells in PBMCs from KI-528 at two different time points. These T cells were detected using TI8/B5201 and TN9-8V/ C1202 tetramers. **C**. Recognition of TN9-8V and TN9-8L peptides by TN9-8V-specific bulk T cells from KI-528. T-cell responses to 721.221-CD4-C1202 cells prepulsed with TN9-8V or TN9-8L peptide were analyzed by ICS, and representative flow cytometric results are shown (left). The relative percent of IFN-γ⁺ cells among CD8⁺ cells was calculated by dividing their response frequency to each peptide by their response frequency to TN9-8V (right). Data are shown as the means and SD of triplicate assays. Statistical analysis was conducted using Student's t-test (two-tailed). \*\*\*$P<0.001$.

viruses in April 2009 (**Fig 6A**). By May 2013, RT135T and RT135I had markedly decreased whereas RT135V predominated (85%). Analysis using two tetramers confirmed TI8-specific T cells in PBMCs from this individual in July 2009, and TN9-8V-specific T cells had emerged by May 2013 (**Figs 6B left and S3C**). These results support the idea that C\*12:02-restricted TN9-8V-specific T cells are elicited after the emergence of 8V mutant virus, which is selected by TI8-specific HLA-B\*52:01-restricted CD8⁺ T cells.

Analysis using HLA-C\*12:02-tetramers specific for the TN9-8V or TN9-8T epitopes revealed that, in July 2009, this individual had HLA-B\*52:01-restricted TI8-specific T cells and HLA-C\*12:02-restricted TN9-8T-specific T cells, but no HLA-C\*12:02-restricted TN9-8V-specific ones (**Figs 6B right and S3C**). This suggests that TN9-8T-specific T cells were elicited by RT135T virus-infected cells, not RT135V virus-infected ones. In contrast, in May 2013,

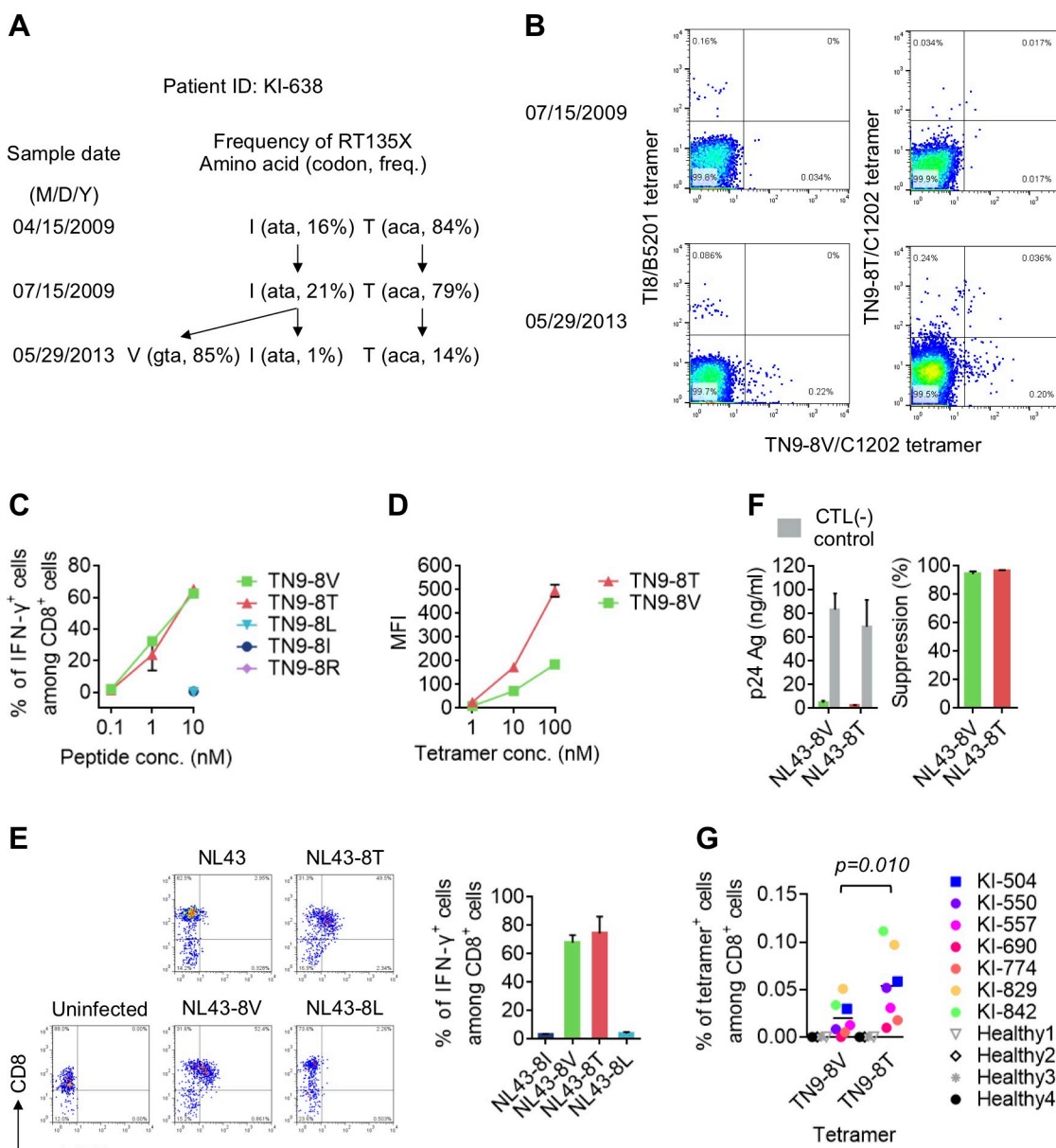

**Fig 6. Viral suppression of RT135T mutant virus by TN9-8T-specific CTL line. A.** Longitudinal sequence analysis at RT135 in an HLA-B*52:01+C*12:02+ individual with HIV-1, KI-638. Frequencies of amino acids at RT135 were determined by both bulk and deep sequencing. Both analyses showed similar results. The amino acids at RT135, their codon usage, and their frequencies are summarized by sampling date. **B.** Detection of TI8-, TN9-8V- and TN9-8T-specific CD8+ T cells in PBMCs from KI-638. These T cells were detected by TN9-8V/C1202 and TI8/B5201 tetramers or both TN9-8T/C1202 and TN9-8V/C1202 tetramers. **C.** Recognition of TN9-8T and its mutant peptides by TN9-8T-specific T cell line derived from KI-638. T-cell responses to 721.221-CD4-C1202 cells prepulsed with TN9-8T or its mutant peptides were analyzed by ICS. **D.** Binding of TN9-8V or TN9-8T tetramer to TN9-8T-specific T cell line. Data are shown as the mean and SD of triplicate assays **E.** Recognition of RT135 mutant virus-infected cells by a TN9-8T-specific T cell line. T-cell responses to 721.221-CD4-C1202 cells infected with NL43 (NL43-8I) or the mutant virus (NL43-8V, -8T or -8L) were analyzed by ICS. The frequencies of p24+ cells among 721.221-CD4-C1202 cells infected with NL43-8I, -8V, -8T, and -8L were 31.8%, 27.0%, 31.8%, and 28.5%, respectively. Representative flow cytometric results (left) and summarized results for the frequency of IFN-γ+ cells among CD8+ cells (right) is shown. **F.** Suppression of viral replication by TN9-8T-specfic CTL line, where E:T ratio was 1:1. The concentration of p24 Ag in the culture supernatants collected at Day 5 was measured by ELISA (left) and the percentage of suppression was calculated (right). **G.** Detection of TN9-8V- and TN9-8T-specific CD8+ T cells in PBMCs from 7 individuals infected with RT135T mutant virus. HLA-C*12:02-restricted TN9-8V and TN9-8T-specific CD8+ T cells were analyzed by using TN9-8T/C1202- or TN9-8V/C1202-tetramer. Statistical analysis was performed using a paired t-test (two-tailed). Data are shown as the means and SD of triplicate assays (**C, D, E, and F**).

three types of HLA-C*12:02-restricted T cells – TN9-8T-specific, TN9-8V-specific, and TN9-8V/8T cross-reactive ones – were observed in this participant (**Fig 6B**). Together this suggests that TN9-8V-specific and TN9-8V/8T cross-reactive T cells were elicited after the emergence of RT135V virus.

To further characterize the TN9-8T-specific T cells found in July 2009, we established a TN9-8T-specific T cell line. This T cell line cross-recognized the TN9-8V peptide to the same level as TN9-8T peptide (**Fig 6C**). The binding affinity of the TN9-8T tetramer to the T cell line was much higher than that of the TN9-8V tetramer (**Fig 6D**). These results indicated that the T cell expressed a T cell receptor (TCR) that could recognize both epitopes, though its affinity for HLA-C*12:02-TN9-8T was higher than its affinity for HLA-C*12:02-TN9-8V. The TN9-8T-specific T cell line also strongly recognized target cells infected with RT135T or RT135V virus (**Fig 6E**), and strongly suppressed the replication of HIV harboring these mutations (**Fig 6F**). Together, these results support the idea that TN9-8T-specific T cells suppressed the replication of RT135T virus in this individual between July 2009 and May 2013.

Finally, to clarify whether TN9-8T-specific T cells are frequently elicited in HLA-B*52:01+C*12:02+ individuals harboring RT135T virus, we stained PBMCs from seven additional HLA-B*52:01+C*12:02+ individuals harboring this mutant virus using HLA-C*12:02/TN9-8T or HLA-C*12:02/TN9-8V tetramers. We detected TN9-8T-specific T cells in four of them (**Figs 6G and S5**), suggesting that TN9-8T-specific T cells are frequently elicited in these individuals. However, the frequency of HLA-C*12:02/TN9-8V tetramer+ T cells was significantly lower than that of HLA-C*12:02/TN9-8T tetramer+ ones (**Figs 6G and S5**). This indicates that TN9-8T-specific T cells are also preferentially elicited in these individuals.

## Discussion

HIV-1 escape mutations reduce CTL recognition of HIV-infected cells by reducing or abrogating epitope processing, HLA binding and/or TCR recognition of the HLA-epitope complex [10,39,40]. Although many escape mutant epitopes cannot elicit specific T cells, some of them can to some extent [41–46]. For example, CTLs specific for two superimposed HLA-A*24:02-restricted epitopes, NefRF10 and NefRW8, select the Nef135F mutation [47]. Whereas the 10-mer epitope having this mutation (RF10-2F) further elicited HLA-A*24:02-restricted CTLs with high affinity TCR specific for RF10-2F, the 8-mer mutant epitope failed to elicit HLA-A*24:02-restricted T cells specific for RW8-2F [48]. We also speculated that immune escape could generate a new epitope recognized by another HLA class I-restricted CTL, which in turn could drive further escape, but such a phenomenon had not been reported to date. Here, we demonstrated that a new HLA-C*12:02-restricted epitope TN9-8V was generated by the RT135V mutation which was selected by HLA-B*52:01-restricted TI8-specific T cells in individuals carrying the HLA-B*52:01-C*12:02 haplotype. Moreover, due to strong linkage disequilibrium between these alleles and the high prevalence (20%) of this haplotype in Japan [34,36], this novel TN9-8V epitope was generated in the Japanese population far more so than in other global populations.

Japanese hemophiliacs acquired HIV-1 strains circulating in North America via contaminated blood products imported from the USA around 1983. To exclude the effect of ongoing HIV-1 domestic transmission on population-level accumulation of escape mutations, we began by studying these individuals. We demonstrated that RT135T and RT135V were significantly enriched among hemophiliacs carrying HLA-B*51:01+ and HLA-B*52:01+, respectively, suggesting that these mutations are selected by HLA-B*51:01- and HLA-B*52:01-restricted TI8-specific T cells, respectively, in this group. Previous studies demonstrated that neither HLA-B*51:01- nor HLA-B*52:01-restricted TI8-specific CTLs could recognize cells infected

with HIV-1 carrying RT135T, RT135L, or RT135R mutations, whereas they could recognize those infected with RT135V mutant virus [2,30,33]. HLA-B*51:01-restricted TI8-specific T cells from individuals harboring wild-type (RT135I) virus suppressed replication of RT135V and wild-type viruses to a similar extent, whereas those from individuals harboring RT135V virus suppressed RT135V mutant virus more readily than wild-type virus in a competitive suppression assay [30]. In contrast, HLA-B*52:01-restricted TI8-specific T cells suppressed replication of wild-type virus more effectively than RT135V virus [33]. These findings together partially explain why HLA-B*52:01-restricted TI8-specific T cells select RT135V more often than HLA-B*51:01-restricted ones. It does not explain why in the Japanese hemophiliac group however, HLA-B*52:01-restricted T cells did not select other RT135 mutants such as RT135T, L or R as was observed in the non-hemophiliac group. Furthermore, although RT135V was enriched in HLA-B*52:01+ hemophiliacs, it was not enriched in HLA-B*52:01+ non-hemophiliacs. This may be explained by the source of transmitted HIV-1: whereas hemophiliacs acquired HIV-1 from North America around 1983, non-hemophiliacs predominantly acquired HIV-1 strains circulating in Japan. Since non-hemophiliac Japanese individuals diagnosed before 1997 predominantly harbored RT135T/L mutations, it is likely that these mutant viruses were initially transmitted and subsequently accumulated in domestically-acquired HIV cases, who were mostly men who have sex with men.

RT135 mutation frequencies have shifted dramatically over the past 20 years in non-hemophiliac Japanese individuals with HIV-1. Among those diagnosed before 1997, RT135T dominated (41%), but its frequency decreased to 25% among those diagnosed between 2010–2015. In contrast, RT135L dominated at this time (44%), whereas its frequency was only 22% among those diagnosed before 1997. HLA-B*51:01+ and HLA-B*51:01-HLA-B*52:01- non-hemophiliac individuals showed similar patterns of change in mutation frequencies over the past 20 years: a decrease in RT135T frequency and an increase in RT135L. This may be explained by viral transmission from HLA-B*52:01+C*12:02+ individuals who selected RT135L but suppressed replication of RT135T. The frequency of RT135T in HLA-B*52:01+C*12:02+ individuals was lower than among HLA-B*51:01+ ones, and comparable to that in HLA-B*51:01-HLA-B*52:01- ones who do not have HLA-B*52:01-restricted TI8-specific/HLA-C*12:02-restricted TN9-8V-specific T cells. This raises the possibility that RT135T is not directly selected by HLA-B*52:01-restricted TI8-specific T cells, or is eliminated by TN9-8T/8V-specific T cells in HLA-B*52:01+C*12:02+ individuals, as outlined in our proposed mechanistic model shown in Fig 7. On the other hand, in British Columbia, Canada, where the frequencies of HLA-B*51:01 and HLA-B*52:01 or HLA-C*12:02 were 11.7% and 1.4%, respectively, amino acid variation at RT135 remained extremely stable over a >15 year period with RT135T consistently representing the dominant mutation (26–28%) [49]. These findings further support the idea that HLA-B*52:01- and/or HLA-C*12:02-mediated immune responses drove the shifts in RT135 mutation frequencies over the last 20 years in Japan.

The frequency of HLA-B*51:01-restricted TI8-specific T cells correlated inversely with pVL whereas that of HLA-B*51:01-restricted T cells specific for three other epitopes did not [29], suggesting that TI8-specific T cells contribute to HLA-B*51:01-restricted HIV-1 control. Consistent with this, HLA-B*51:01 was previously a protective allele in Japanese hemophiliacs [29] and other populations [26,50]. However, this allele is no longer protective in non-hemophiliac Japanese individuals [31,32]. Since 80% of HLA-B*51:01+ Japanese individuals harbor RT135T/L, it is unlikely that HLA-B*51:01+ TI8-specific T cells can control replication of RT135T/L mutant virus in these persons. Thus, accumulation of RT135T/L at the population level critically compromised HLA-B*51:01-mediated control of HIV-1. Instead, HLA-B*52:01-C*12:02 represents a protective haplotype in non-hemophiliac Japanese individuals [32]. Consistent with this, a genome-wide association study of HIV-1 control in White

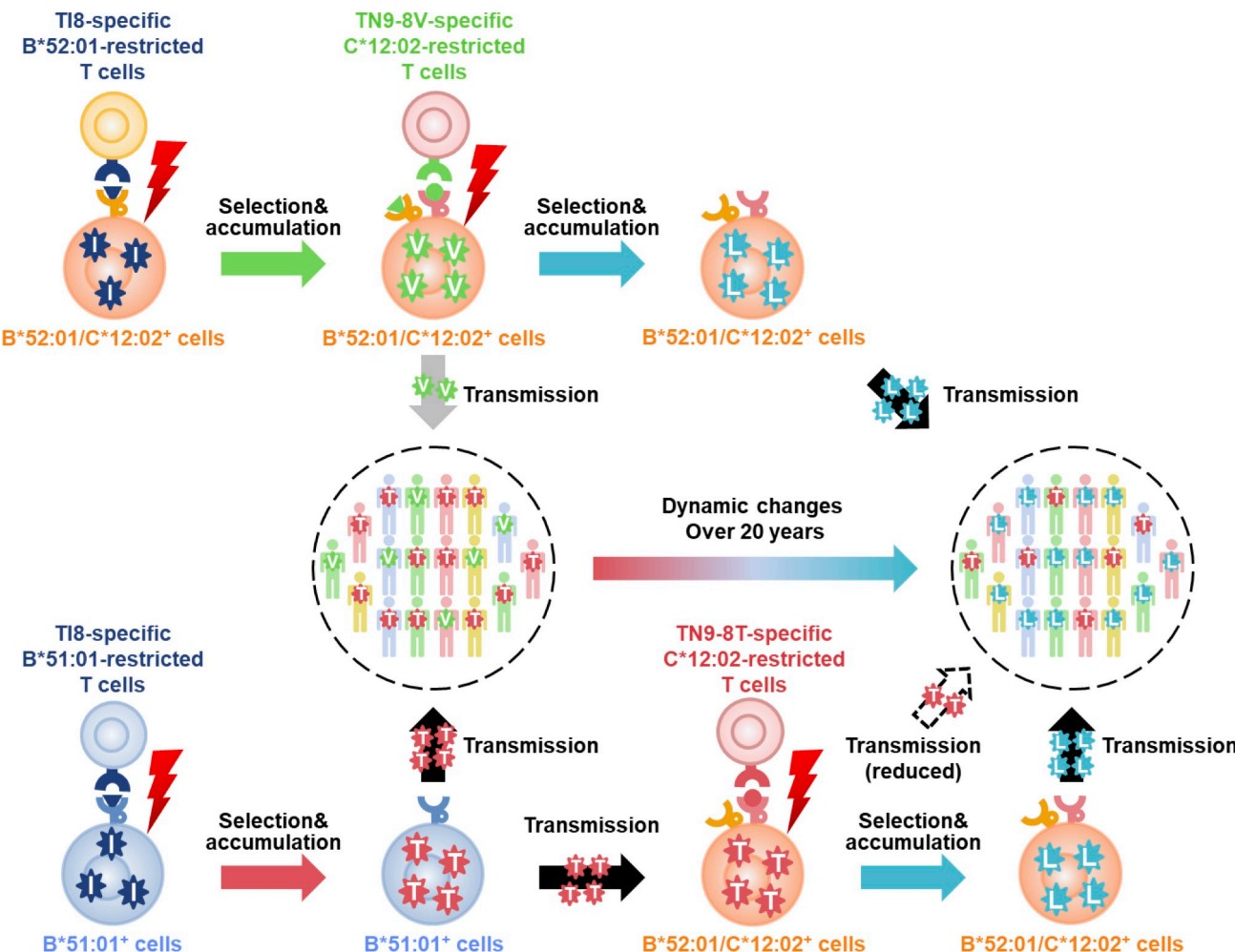

**Fig 7. Proposed mechanism for coevolution of HIV-1 with HIV-1-specific T cells.** RT135V and RT135T viruses were selected by TI8-specific HLA-B*52:01-restricted and TI8-specific HLA-B*51:01-restricted T cells, respectively. In the case of selection of RT135V virus in HLA-B*52:01+HLA-C*12:02+ persons or transmission of this mutant virus to them, HLA-C*12:02-restricted TN9-8V-specific T cells were subsequently elicited, which led to the selection of RT135L virus rather than RT135T, since these T cells can suppress replication of RT135T virus but not RT135L. In the case of RT135T transmission to HLA-B*52:01+HLA-C*12:02+ individuals, HLA-C*12:02-restricted TN9-8T-specific T cells were subsequently elicited, which effectively suppressed replication of this mutant virus. Since RT135L virus cannot be recognized by HLA-C*12:02-restricted TN9-8T-specific T cells, it can be selected in HLA-B*52:01+HLA-C*12:02+ individuals infected with RT135T virus.

(Caucasian) and Black individuals identified HLA-B*52:01, but not HLA-B*51:01, as a protective allele together with HLA-B*57 and HLA-B*27 [51]. Previous studies described four HLA-B*52:01-restricted and two HLA-C*12:02-restricted protective epitopes [35,52] but TI8 was not among these. In contrast, our recent study of HLA-B*52:01+ TI8 responders diagnosed from 2011 to 2017 in Japan reported good clinical outcome in these individuals [53], suggesting that HLA-B*52:01-restricted TI8-specific T cells may have the ability to control HIV-1 in individuals who acquired the virus in more recent years. The role of TN9-8V-specific T cells in HIV-1 control *in vivo* remains unclear, and merits further study.

In summary, we demonstrated dynamic shifts in HIV-1 RT135 mutation frequency over the past two decades in Japan, where RT135T frequency significantly decreased among non-hemophiliacs with HIV-1, and RT135L significantly increased. We further describe a plausible mechanism for these shifts, summarized in **Fig 7**. Specifically, we describe novel T cell responses restricted by the most common HLA-B/HLA-C allele haplotype expressed in Japan,

and demonstrate that these likely contributed to dynamic population-level changes in RT135 mutation frequencies in Japan that are not mirrored in other parts of the globe. Our observations underscore HIV-1's dynamic ability to co-evolve with virus-specific T cells in a host population-specific manner.

## Methods

### Ethics statement

This study was approved by the ethics committee of Kumamoto University (RINRI-1340 and GENOME-342) and the National Center for Global Health and Medicine (NCGM-A-000172-01). Written informed consent was obtained from all individuals according to the Declaration of Helsinki.

### Participants

Plasma and peripheral blood mononuclear cells (PBMCs) were separated from whole blood. HLA genotypes were determined by the HLA Foundation Laboratory, Kyoto, Japan. Plasma and PBMCs from 23 HLA-B*52:01+HLA-C*12:02+ individuals with chronic HIV-1 subtype B were characterized with respect to HIV-1 sequence and CTL response. HIV-1 sequences from 95 Japanese hemophiliac patients, who acquired HIV-1 subtype B via contaminated blood products imported from the USA around 1983, were analyzed for HIV-1 amino acid variation at RT135. For comparison, amino acid frequencies at RT135 derived from 1,478 Japanese individuals with subtype B HIV-1 enrolled in NCGM from before 1997 to 2015, and 6,258 British Columbians with subtype B HIV-1 collected between 2000 and 2015 [49] were also analyzed.

### Cells

C1R cells expressing HLA-B*52:01 (C1R-B5201) and HLA-C*12:02 (C1R-C1202), and 721.221 cells expressing CD4 and HLA-C*12:02 (721.221-CD4-C1202) were previously generated [33,54]. These cells were cultured in RPMI-1640 medium (Invitrogen) containing with 5–10% fetal bovine serum (R5-R10) and 0.15 mg/ml hygromycin B. RMA-S-C1202 cells, a TAP deficient mouse-derived cell line that expresses HLA-C*12:02, were previously generated and cultured in R10 containing 0.15 mg/ml hygromycin B [55]. MAGIC-5 cells (CCR5-transfected HeLa-CD4/LTR-β-gal cells) were cultured in R10 as described previously [56].

### Induction of peptide-specific bulk T cells

PBMCs from HLA-B*52:01+ and HLA-C*12:02+ expressing individual with chronic HIV-1, KI-793, were stimulated with KN11 or KN11-10V peptide (1 μM) in R10 supplemented with 200 U/ml human recombinant interleukin-2 (rIL-2) to induce peptide-specific bulk T cells for 14 days. After induction, bulk T cells were examined for IFN-γ production using an intracellular cytokine staining (ICS) assay.

### ICS assay

C1R-C1202 or 721.221-CD4-C1202 cells were pre-incubated with or without their cognate peptide(s) at concentrations from 0.1 to 10 nM at 37˚C for 1 hr, and washed twice with R10. Cultured bulk T cells or established CTL lines were incubated with peptide-pulsed or virus-infected 721.221-CD4-C1202 cells in R10 containing Brefeldin A (10 μg/ml) in a 96-U plate (Nunc) at 37˚C for 4 hrs. Subsequently, the cells were stained with allophycocyanin (APC)-conjugated anti-CD8 monoclonal antibody (mAb) (Dako) and 7-amino-actinomycin D (7-AAD) (BD Bioscience) at 4˚C for 30 min, after which the cells were fixed with 4%

paraformaldehyde solution and permeabilized with permeabilization buffer (0.1% saponin and 5% FBS in phosphate-buffered saline) at 4°C for 10 min. Thereafter the cells were stained with fluorescein isothiocyanate (FITC)-conjugated anti-IFN-γ mAb (BD Biosciences) at room temperature for 30 min and then washed twice with the permeabilization buffer. The percentage of CD8[+] cells producing IFN-γ was analyzed by flow cytometry (FACS Canto II).

### Generation of TN9-8V- and TN9-8T-specific CTL lines

TN9-8V-specific CTL lines were generated from PBMCs of three chronically HIV-1-infected individuals, KI-069, -661 and -793, while TN9-8T-specific CTL lines were generated from PBMCs from chronically HIV-1-infected individual KI-638. PBMCs were stained with TN9-8V/C1202- or TN9-8T/C1202-tetramer at 37°C for 30 min and washed twice with R10, after which they were stained with FITC-conjugated anti-CD3, Pacific blue-conjugated CD8 and 7-Aminoactinomycin D (7AAD) viability dye at 4°C for 30 min and washed twice with R10. After staining, TN9-8V- or TN9-8T-specific CD8[+] T cells were sorted by FACSAria into a 96-well U-bottom plate and cultured with irradiated feeder and stimulator cells in cloning medium containing 100 nM of TN9-8V or TN9-8T peptide. Half of the medium was replaced every 2–3 days with fresh medium without peptide. After 14–21 days in culture, the growing cells were tested for their ability to recognize TN9-8V or TN9-8T peptide by ICS assay.

### HIV-1 clones

NL43-RT135X mutant viruses were previously generated by site-directed mutagenesis [2].

### Generation of HLA-peptide tetrameric complexes

HLA class I-peptide tetrameric complexes (tetramers) were synthesized as previously described [57]. TI8/B5201 monomer complex was previously generated [33]. TN9-8V or TN9-8T peptide was added to the refolding solution containing the biotinylation-sequence-tagged extracellular domain of HLA-C*12:02 molecule and β2 microglobulin. The purified monomer complexes were mixed with Phycoerythrin (PE)- or APC labeled streptavidin (Molecular Probes) at a molar ratio of 4:1.

### Tetramer binding assay

PBMCs and the TN9-8T-specific CTL line were stained with PE or APC-conjugated tetramer at 100 nM and 1–100 nM, respectively at 37°C for 30 min. After washing twice with R10, the cells were stained with FITC-conjugated anti-CD3 mAb, Pacific blue-conjugated anti-CD8 mAb and 7-AAD at 4°C for 30 min. Thereafter, the cells were washed twice with R10 and then analyzed by flow cytometry (FACS Canto II).

### Viral replication suppression assay

The ability of TN9-8V-specific and TN9-8T-specific CTL lines to suppress HIV-1 replication was examined as previously described [58]. Activated CD4[+] T cells ($1 \times 10^6$ or $5 \times 10^5$ cells) were isolated from PBMCs of healthy HLA-B*52:01[+] C*12:02[+] donors and incubated with a given HIV-1 clone at 37°C for 6 hrs. After washing twice with R10, the cells ($1 \times 10^4$/well) were co-cultured with TN9-8V- or TN9-8T-specific CTL lines at E:T ratios of 1:1 in culture media containing IL2. From day 3 to day 7 post-infection, 30 μl of culture supernatant was collected and replaced with fresh medium. The concentration of p24 Ag was measured using an enzyme-linked immunosorbent assay (ELISA) (ZeptoMetrix or Rimco). Percent suppression was

calculated as follows: (concentration of p24 Ag without the CTLs—concentration of p24 Ag with the CTLs)/concentration of p24 Ag without the CTLs x 100.

## Sequence of autologous virus

For longitudinal HIV-1 sequence analysis, data were obtained as follows. Viral RNA was extracted from plasma using a QIAamp MinElute virus spin kit (QIAGEN). cDNA was synthesized from the RNA with SuperScript III first-strand kit (Invitrogen) and then the Pol region was amplified by nested PCR with Taq DNA polymerase (Promega). DNA sequencing was performed using a BigDye Terminator cycle sequencing kit (Applied Biosystems) on an ABI PRISM genetic analyzer.

To determine the frequencies of minority RT135 variants in studied samples, deep sequencing was performed using Miseq (Illumina). PCR products were purified by PCR-cleanup kit (SIGMA). Tagmentation and normalization were performed using the Nextera XT sample preparation kit (Illumina). Normalized samples were pooled and loaded into Miseq reagent kit v3 600 cycles and sequenced following the manufacturer's instructions. Miseq reads were quality trimmed using Sickle (https://github.com/najoshi/sickle). The reads with high-quality scores (Q>30) were mapped to the HIV-1 NL4-3 reference genome (GenBank: AF324493) using Bowtie2. After sorting and converting using samtools, mapped reads were visualized using the Integrative Genomics Viewer (IGV) after which amino acid frequencies were calculated.

## HLA class I stabilization assay

Peptide-binding activity to HLA-C*12:02 was examined using RMA-S-C1202 cells as previously described [59]. Briefly, RMA-S-C1202 cells were cultured at 26˚C for 16 h and then pulsed with peptides at concentrations of 1 to 100 μM at 26˚C for 1 h. Subsequently, the cells were incubated at 37˚C for 3 h, followed by staining with anti-HLA-Bw6 antibody (SFR8-B6) which cross-binds to HLA-C [60] and FITC-conjugated sheep anti-mouse IgG (Jackson Immuno Research). The MFI was measured by flow cytometry (FACS Canto II). Relative MFI was calculated by subtracting the MFI of peptide-unpulsed cells from that of the peptide-pulsed ones.

## Statistical analysis

The frequency of the amino acids between HLA$^+$ and HLA$^-$ individuals was analyzed by using Fisher's exact test. Multiple tests were addressed using $q$ values, the $p$-value analogue of the false discovery rate. A significance threshold of $q$ <0.2 was employed. The differences in the peptide binding affinities (Fig 3B) or in the T-cell responses (Fig 5C) between TN9-8V and other peptides were statistically analyzed by performing the two-tailed unpaired $t$ test. The difference between tetramer$^+$ T-cell frequencies was compared using the paired $t$ test (Fig 6G). All tests were two-tailed and $p$ values of <0.05 were considered significant.

The association between V, T, R, M, L, K, I frequencies across the four periods was assessed using the Cochran–Mantel–Haenszel test, where the mutations were treated as nominal variables and the four periods were treated as ordered categorical data (Fig 4A). Analyses were performed overall, and stratified by allele carriage. Statistical analyses were performed using SAS (SAS Institute), version 9.4.

## Supporting information

**S1 Fig. Nucleotide substitution at RT135 in Japanese individuals with chronic HIV-1, Related to Fig 1.** Codon usages for I, V, and T are shown, along with their frequencies observed in Japanese individuals with chronic HIV-1 (Hemo; 95 hemophiliacs shown in Fig

1A, Non-hemo; 83 non-hemophiliac individuals shown in Fig 1B and S1 Table).
(TIF)

**S2 Fig. Identification of a novel HLA-C*12:02-restricted CTL epitope including RT135, Related to Fig 2A.** A. T-cell responses to 11-mer overlapping Pol peptides containing RT135. T-cell responses of PBMCs from an HLA-B*52:01+C*12:02+ individual (KI-793) to five 11-mer overlapping Pol peptides containing RT135 position and Pol peptide cocktail 15 including the 5 overlapping peptides were analyzed at a concentration of 100 nM by ELISPOT assay. B. Identification of HLA-restriction of the response to the Pol 11–141 (KN11) peptide. IFN-γ production from KN11-induced bulk T cells stimulated with C1R cells expressing either HLA-B*52:01 or -C*12:02 molecule pre-pulsed with the KN11 peptide at concentration of 100 nM was analyzed by ICS assay. C. Comparison of induction efficiency between bulk T cells induced with the KN11 peptide and those with KN11-10V mutant one. IFN-γ production from KN11-induced or KN11-10V-induced bulk T cells stimulated with C1R-C1202 cells pre-pulsed with the KN11 or KN11-10V peptide was analyzed by ICS assay. Relative % of IFN-γ+ cells among CD8+ T cells was calculated as follows: % of IFN-γ+ cells with peptide–% of IFN-γ+ cells without peptide.
(TIF)

**S3 Fig. Negative controls of ex vivo tetramer staining assay, Related to Figs 2B, 5B and 6B.** A. Staining of KI-793 PBMCs without tetramer, related to Fig 2B. B. Staining of KI-528 PBMCs without tetramers, related to Fig 5B. C. Staining of KI-638 PBMCs without tetramers (upper) and staining of PBMCs derived from a healthy donor with tetramers (bottom), related to Fig 6B.
(TIF)

**S4 Fig. Recognition of virus-infected cells by TN9-8V-specific T cell lines, Related to Fig 3C.** Responses by TN9-8V-specific CTL lines were established from 3 HLA-B*52:01+C*12:02+ individuals with chronic HIV-1. The ability of these T cells to recognize 721.221-CD4-C1202 cells infected with NL43, or NL43-RT135X mutant viruses was analyzed by ICS assay. Frequency of IFN-γ+ cells among CD8+ cells was indicated.
(TIF)

**S5 Fig. *Ex vivo* detection of TN9-8T-specific CD8+ T cells, Related to Fig 6G.** PBMCs from seven HLA-B*52:01+ C*12:02+ individuals harboring HIV-1 RT135T virus were stained with TN9-8V/C1202 tetramer or TN9-8T one at concentration of 100 nM. Representative cases corresponding to Fig 6G were shown. Frequency of tetramer+ cells among CD3+CD8+ T cells is indicated in red.
(TIF)

**S1 Table. Codon usages of amino acids at RT135 observed in Japanese hemophiliacs and non-hemophiliacs with chronic HIV-1**
(DOCX)

**S2 Table. Statistical analysis using Cochran-Mantel-Haenszel test on the association between the frequency of 7 amino acids and the four periods.**
(DOCX)

## Acknowledgments

The authors thank Jeffrey Joy, Julio Montaner and the BC Centre for Excellence in HIV/AIDS laboratory for data access and support.

## Author Contributions

**Conceptualization:** Masafumi Takiguchi.

**Data curation:** Tomohiro Akahoshi, Hiroyuki Gatanaga, Takayuki Chikata.

**Formal analysis:** Naoki Ishizuka.

**Funding acquisition:** Tomohiro Akahoshi, Zabrina L. Brumme, Masafumi Takiguchi.

**Investigation:** Tomohiro Akahoshi, Hiroyuki Gatanaga, Nozomi Kuse, Takayuki Chikata, Madoka Koyanagi, Chanson J. Brumme, Hayato Murakoshi, Zabrina L. Brumme.

**Project administration:** Masafumi Takiguchi.

**Resources:** Tomohiro Akahoshi, Hiroyuki Gatanaga, Nozomi Kuse, Chanson J. Brumme, Zabrina L. Brumme, Shinichi Oka.

**Supervision:** Masafumi Takiguchi.

**Validation:** Masafumi Takiguchi.

**Visualization:** Tomohiro Akahoshi, Nozomi Kuse, Masafumi Takiguchi.

**Writing – original draft:** Tomohiro Akahoshi, Zabrina L. Brumme, Masafumi Takiguchi.

**Writing – review & editing:** Tomohiro Akahoshi, Hiroyuki Gatanaga, Nozomi Kuse, Zabrina L. Brumme, Masafumi Takiguchi.

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
