## [Decision Letter · Decision Letter 0]

23 Oct 2020

Dear Dr. Takiguchi,

Thank you very much for submitting your manuscript "T-cell responses to sequentially emerging viral escape mutants shape long-term HIV-1 population dynamics" for consideration at PLOS Pathogens. As with all papers reviewed by the journal, your manuscript was reviewed by members of the editorial board and by several independent reviewers. The reviewers appreciated the attention to an important topic. Based on the reviews, we are likely to accept this manuscript for publication, providing that you modify the manuscript according to the review recommendations.

The reviewers make a number of suggestions for clarifying the manuscript of this paper. Specifically, I would ask you to shorten the text significantly, removing unnecessary comment and discussion (cf. Reviewer 2), and most important, clarify the novel contributions that your results make to the extensive existing literature on the impact of T cells on HIV-1 sequences and population dynamics.

Sincerely,

Charles R. M. Bangham

Associate Editor

PLOS Pathogens

Richard Koup

Section Editor

PLOS Pathogens

Kasturi Haldar

Editor-in-Chief

PLOS Pathogens

orcid.org/0000-0001-5065-158X

Michael Malim

Editor-in-Chief

PLOS Pathogens

orcid.org/0000-0002-7699-2064

The reviewers make a number of suggestions for clarifying the manuscript of this paper. Specifically, I would ask you to shorten the text significantly, removing unnecessary comment and discussion (cf. Reviewer 2), and most important, clarify the novel contributions that your results make to the extensive existing literature on the impact of T cells on HIV-1 sequences and population dynamics.

Reviewer Comments (if any, and for reference):

Reviewer's Responses to Questions

**Part I - Summary**

Reviewer #1: In the manuscript entitled “T-cell responses to sequentially emerging viral escape mutants shape long-term HIV-1 population dynamics”, Akahoshi et al. demonstrated HIV-1 coevolved with multiple virus-specific T cells in a host population specific manner, indicating dynamic HIV sequence changes at population level. Specifically, HIV-1 reverse transcriptase codon 135 (RT135) mutant RT135V was selected by HLA-B*52:01-restricted T cells and created a new epitope restricted by HLA-C*12:02. The authors identified novel HLA-C*12:02-restricted T cell epitopes at RT135 and investigated the role of these epitope-specific T cells in selecting RT135 mutants among HLA- B*52:01+ C*12:02+ population. Future, the authors investigated dynamic changes of RT135 mutations over 20 years, in the context of HIV long-term coevolution with multiple virus-specific T cells. The virus-specific T cells were validated at the functional level using cytokine staining and tetramer staining, not just at the sequence level alone. Overall, this is a very interesting and elegant study tracking viral-host interactions and the immune selection pressure at a population level at the scale of decades.

Reviewer #2: In the study by Akahoshi et al, the authors present a study consisting of population-level virology and immunology. They demonstrate HLA associations and viral evolution patterns of HIV-1 epitope escape variants. This study compares viral and immune features between 95 hemophiliacs and 1,478 non-hemophiliac HIV-1-infected men. The inclusion of hemophiliacs is very clever, as it represents a "control" population presumably initially infected with WT virus. This study is well-controlled, the experiments are properly performed, and it presents some very interesting findings that are of relevance to the field. Despite this, there are several issues, mostly associated with narrative choices and style, that detract from the study.

Reviewer #3: Akahoshi et al present a manuscript with the title: T-cell responses to sequentially emerging viral escape mutants shape long-term HIV-1 population dynamics.

They show that HIV-1 strains harboring escape mutations selected by virus-specific cytotoxic T lymphocytes accumulate at the population level. In this context they investigated population-level changes for a single epitope at HIV-1 Reverse Transcriptase codon 135 (RT135), which is under pressure by T cells restricted by two different HLA-B alleles that are frequent in Japan. They found that RT135V selection by HLA-B*52:01-restricted T-cells resulted in the emergence of a new epitope restricted by HLA-C*12:02. Their results suggest multiple virus-specific T cells as dynamic drivers of population-level HIV-1 evolution.

While the study is of interested for the specific understanding of HIV evolution under immune pressure its design is extremely narrow. The authors study the evolution and immune pressure for only one T cell epitope (RT135). The HLA alleles they focus on are especially frequent in Japan but not, for example, in Canada. Thus, the study reports a very specific finding, which is probably not of interest for a wide range of scientists in the field of infectious diseases, but rather for HIV researchers only. It would therefore be more appropriate for a specialized HIV journal. Along these lines, the generalized statement at the end of the abstract:

“Together, our observations indicate that host population-specific T-cell responses to sequentially emerging viral escape mutants can shape long-term HIV-1 population dynamics, thus demonstrating HIV-1 coevolution with multiple virus-specific T cells.“

sounds interesting and meaningful, but has already been shown by several previous publications. Thus, the current study does not provide findings that are novel and outstanding enough to justify publication in a journal that is reflecting the full breadth of infectious diseases, like PLoS Pathogens.

**Part II – Major Issues: Key Experiments Required for Acceptance**

Reviewer #1: None.

Reviewer #2: No new experiments are recommended

Reviewer #3: None

**Part III – Minor Issues: Editorial and Data Presentation Modifications**

Reviewer #1: 1. The authors may more explicitly point out hemophiliac (figure 1a) and non-hemophiliac (figure 1d) individuals in the figure and figure legend (figure 1d).

2. Given that tetramer staining may be tricky due to the low frequency of positive cells, do the authors have negative control plots for figure 1b, figure 2b, figure 5b, and figure 6b?

3. Figure 7 legend should include some brief description of the co-evolution of HIV-1 with virus-specific T cells.

4. The authors may discuss the impact of dynamic population-level changes of HIV-1 sequences and how observed HIV RT135 mutations affect T cell-mediate virus control.

Reviewer #2: 1. The manuscript in general is very dense, wordy, and difficult to follow, with diffuse goals. It could benefit from a very clear goal. Much of the excess verbiage should be trimmed to focus only on the key messages that are integral to the main point of the paper.

2. Similarly, the abstract should be clear and concise, and convey a central message

3. Much of the introduction could be omitted, moved to the discussion, or condensed

4. In some cases, figure panels and comparisons can be combined or rearranged. For example, Figures 1A and D make more sense to be discussed back to back. Figure 1B and C seem superfluous and could realistically be moved to supplementary.

Reviewer #3: None

PLOS authors have the option to publish the peer review history of their article (what does this mean?). If published, this will include your full peer review and any attached files.

Reviewer #1: No

Reviewer #2: No

Reviewer #3: No
---

## [Editor Report · Decision Letter 1]

18 Nov 2020

Dear Dr. Takiguchi,

We are pleased to inform you that your manuscript 'T-cell responses to sequentially emerging viral escape mutants shape long-term HIV-1 population dynamics' has been provisionally accepted for publication in PLOS Pathogens.

Best regards,

Charles R. M. Bangham

Associate Editor

PLOS Pathogens

Richard Koup

Section Editor

PLOS Pathogens

Kasturi Haldar

Editor-in-Chief

PLOS Pathogens

orcid.org/0000-0001-5065-158X

Michael Malim

Editor-in-Chief

PLOS Pathogens

orcid.org/0000-0002-7699-2064
---

## [Editor Report · Acceptance letter]

22 Dec 2020

Dear Dr. Takiguchi,

We are delighted to inform you that your manuscript, "T-cell responses to sequentially emerging viral escape mutants shape long-term HIV-1 population dynamics," has been formally accepted for publication in PLOS Pathogens.

Best regards,

Kasturi Haldar

Editor-in-Chief

PLOS Pathogens

orcid.org/0000-0001-5065-158X

Michael Malim

Editor-in-Chief

PLOS Pathogens

orcid.org/0000-0002-7699-2064